# Phylogeny, Virulence, and Antimicrobial Resistance Gene Profiles of *Enterococcus faecium* Isolated from Australian Feedlot Cattle and Their Significance to Public and Environmental Health

**DOI:** 10.3390/antibiotics12071122

**Published:** 2023-06-28

**Authors:** Yohannes E. Messele, Darren J. Trott, Mauida F. Hasoon, Tania Veltman, Joe P. McMeniman, Stephen P. Kidd, Kiro R. Petrovski, Wai Y. Low

**Affiliations:** 1The Davies Livestock Research Centre, School of Animal and Veterinary Sciences, University of Adelaide, Adelaide, SA 5371, Australia; yohannes.messele@adelaide.edu.au (Y.E.M.);; 2The Australian Centre for Antimicrobial Resistance Ecology, University of Adelaide, Adelaide, SA 5005, Australia; 3Meat & Livestock Australia, Level 1, 40 Mount Street, North Sydney, NSW 2060, Australia; 4Research Centre for Infectious Disease, School of Biological Sciences, University of Adelaide, Adelaide, SA 5005, Australia

**Keywords:** phylogeny, multidrug resistance, vancomycin, virulence factor

## Abstract

The extent of similarity between *E. faecium* strains found in healthy feedlot beef cattle and those causing extraintestinal infections in humans is not yet fully understood. This study used whole-genome sequencing to analyse the antimicrobial resistance profile of *E. faecium* isolated from beef cattle (n = 59) at a single feedlot and compared them to previously reported Australian isolates obtained from pig (n = 60) and meat chicken caecal samples (n = 8), as well as human sepsis cases (n = 302). The *E. faecium* isolated from beef cattle and other food animal sources neither carried *vanA*/*vanB* responsible for vancomycin nor possessed *gyrA*/*parC* and *liaR*/*liaS* gene mutations associated with high-level fluoroquinolone and daptomycin resistance, respectively. A small proportion (7.6%) of human isolates clustered with beef cattle and pig isolates, including a few isolates belonging to the same sequence types ST22 (one beef cattle, one pig, and two human isolates), ST32 (eight beef cattle and one human isolate), and ST327 (two beef cattle and one human isolate), suggesting common origins. This provides further evidence that these clonal lineages may have broader host range but are unrelated to the typical hospital-adapted human strains belonging to clonal complex 17, significant proportions of which contain *vanA*/*vanB* and *liaR*/*liaS*. Additionally, none of the human isolates belonging to these STs contained resistance genes to WHO critically important antimicrobials. The results confirm that most *E. faecium* isolated from beef cattle in this study do not pose a significant risk for resistance to critically important antimicrobials and are not associated with current human septic infections.

## 1. Introduction

Enterococci are ubiquitous Gram-positive commensal bacteria that encompass more than 50 species found in various environments, such as the gastrointestinal tract of animals and humans, as well as the hospital environment [1]. Some *Enterococcus* spp. such as *Enterococcus faecalis* strain Symbioflor 1 are used as probiotics to treat diarrhoea, improve immunity, and provide other health benefits [2,3]. However, other *Enterococcus* spp. such as *Enterococcus faecium* are among the most important causes of hospital-acquired infections in humans, including endocarditis, sepsis, urinary tract, and central nervous system infection [1,4]. More than 90% of enterococcal infections identified in patients with bacteraemia are caused by either *E. faecium* or *Enterococcus faecalis* [5]. In additional to their ability to form biofilms, *E. faecium* and *E. faecalis* possess additional virulence factors that have contributed to their pathogenesis and clinical significance [6]. Some of these virulence factors include the production of enzymes and toxins which allow the bacteria to adhere to and colonize host tissues, modulate host immunity, and increase the severity of infection [7]. 

*E. faecium* are also intrinsically resistant to various antimicrobial classes, including aminoglycosides, cephalosporins, and trimethoprim-sulfamethoxazole [1,8,9]. *E. faecium* is known for its efficiency in recruiting and exchanging AMR determinants through mutation or horizontal transfer of ARGs located on mobile genetic elements (MGEs), which can be readily transferred between *E. faecium* lineages [10]. *E. faecium* is also capable of acquiring plasmid-mediated resistance determinants from other *Enterococcus* species [11]. Moreover, the transfer of vancomycin resistance from *E. faecium* to methicillin-resistant *Staphylococcus aureus* (MRSA) via broad host range plasmids is a major concern [12]. Plasmids found in enterococci are categorized based on their rep (replication) families, which have either narrow or broad host ranges. Rep-family plasmids such as rep2, rep3, rep5, rep8, rep9, rep11, rep12, rep14, rep15, rep16, rep18, and rep19 are narrow host range, whereas others, like rep1, rep4, rep6, rep7, rep10, and rep13 are considered broad host range [13]. The presence of plasmids in enterococci is extensive, as demonstrated by the detection of at least one and up to seven plasmids in 94.6% of *E. faecium* isolates [14]. Multidrug-resistant *E. faecium* is the most common cause of hospital-acquired infections compared to other species of enterococci [15]. In addition, many human *E. faecium* isolates have developed resistance to last-resort antimicrobials, such as daptomycin, linezolid, tigecycline, and vancomycin, which could have dire consequences for public health in the future [8,9,16,17,18]. Among clinical *E. faecium* isolates worldwide, the *vanA* and *vanB* vancomycin resistance genotypes are the most prevalent ARG variants [19]. 

Whilst some genetic similarity has been found among enterococci isolated from animals and those causing human infections [20], other studies have reported that animal-derived and human-derived enterococci are genetically distinct populations [21,22]. However, there is limited information on the host-species specificity of individual *E. faecium* sequence types (STs), together with their correlation with specific hospital-infection-associated AMR determinants and virulence factors. Australia represents a unique study site to explore the host specificity of animal-origin *E. faecium* isolates given its isolation, quarantine restrictions including a ban on importation of live food-producing animals, and conservative regulation of antimicrobial use in both human and animal health [23]. As a case in point, the use of antimicrobials in livestock production is highly regulated in Australia. For example, Australian legislation prohibits the use of fourth-generation cephalosporins, colistin, fluoroquinolones, and gentamicin in food-producing animals [24,25]. As a result, a low prevalence of AMR has been reported in enterococci isolated from different food-producing animals in Australia compared to other countries [26,27,28]. 

In our previous study, we investigated the frequency of AMR in enterococci from faecal samples collected at the entrance to and exit from a beef feedlot in South Australia [29]. While the prevalence of AMR to individual antimicrobials remained largely static between entry and exit, there was a dramatic shift in *Enterococcus* spp. prevalence, with *E. faecium* infrequently isolated from entry samples (9/104; 8.7%), but then becoming the most prevalent species in exit samples (117/144; 81.3%). The interplay between diet, management practices, and microbial ecology is a potential factor that could contribute to the observed shift in *Enterococcus* spp. [30,31]. This shift in *Enterococcus* prevalence is noteworthy as it highlights the importance of monitoring alterations in microbial communities, where these changes can have implications for both animal health and food safety. Further genomic interrogation is now required to determine the genetic similarity of these cattle-origin *E. faecium* isolates compared to those detected in other food animals and humans in Australia as well as to examine capacity for cross-host species transmission and potential spread via the food chain or the environment. Therefore, this study aimed to conduct further genetic analysis on the beef cattle *E. faecium* isolates through comparing them to isolates obtained from healthy meat chickens and pigs at slaughter as well as human clinical blood sepsis isolates in Australia described in previous studies [5,29]. 

## 2. Results

### 2.1. Distribution of MLST Genotypes and Virulence Factors

Overall, 59 strains of *E. faecium* from beef cattle were analysed via WGS and compared with commensal isolates from healthy slaughter age pigs (n = 60), meat chickens (n = 8), and a large collection of human clinical sepsis isolates (n = 302). SNP analysis of the beef cattle, chicken, and pig isolates confirmed that they formed two main clades (Clade 1 and Clade 2). Clade 2 was subsequently divided into two subclades (Clade 2a and Clade 2b). Clade 2b contained a notable monophyletic subclade (Clade 2ba) that was composed entirely of more closely related human isolates (Figure 1). The predominant animal isolate clades also contained a small proportion of human isolates (23/302; 7.3%) including five isolates each located within the beef-cattle-isolate-dominated Clade 1 and the pig-isolate-dominated Clade 2a (Appendix A). Among beef cattle isolates, 15 distinct sequence types (STs) were identified via MLST, the most common being ST798 in Clade 1 (9/59; 15.2%), followed by ST32 in Clade 2b and ST1036 in Clade 1 (8/59; 13.5% each). By comparison, the most common STs among pig and human isolates were ST5 in Clade 2a (6/60; 10.0%) and ST796 in Clade 2ba (47/302; 15.6%), respectively. Three *E. faecium* STs contained both animal and human isolates. These included ST32, which comprised eight beef cattle isolates and one human isolate; ST22, which comprised one beef cattle isolate, one pig isolate, and two human isolates; and ST327, which comprised two beef cattle isolates and one human isolate. 

Out of the eight *E. faecium* virulence genes identified in this study, *acm* (collagen-binding) and *efaAfm* (adherence) were most prevalent across all sources of isolates. Among the beef cattle isolates, *acm* was found in 56/59 (95.0%) and *efaAfm* in 58/59 (98.3%) (Figure 1). The remaining *E. faecium* virulence genes were exclusively found in human-source isolates in Clade 2ba. These included *agg* (adherence, 2/302; 0.7%), *cylA* (cytolysin, 1/302; 0.3%), *cylL* (cytolysin, 2/302; 0.7%), *cylM* (cytolysin, 1/302; 0.3%), *espfm* (surface protein, 55/302; 18.2%), and *hylEfm* (hyaluronidase 82/302; 27.1%). The human *E. faecium* isolates located in STs 32, 22, and 372 also did not contain *espfm* and *hylEfm* markers typically exclusive to hospital-associated strains, but carried both *acm* and *efaAfm*.

### 2.2. Antimicrobial Resistance Genes Including Those Encoding Resistance to Critically Important Antimicrobials

Heat maps were generated based on the identification of genes from the ResFinder and CARD databases (Figure 2). Overall, 38 ARGs responsible for encoding resistance to aminoglycosides (*aadA5, aadD, aph(2”)-Ia, aph(2”)-Ie, aph(3’)-III, aac(6’)-aph(2”), aac(6’)-Ii, ant(6)-Ia, ant(9)-Ia*), ß-lactams (*blaZ, pbp5*), fluoroquinolones (*efmA, gyrA, parC*), folate pathway inhibitor (*dfrG*), glycopeptides (*vanA, vanB*), lipopeptides (*cls, liaR, liaS*), macrolides/lincosamides, and/or streptogramins (MLS) (*eatAv, erm(A), erm(B), erm(T), lnu(B), lnu(G), lsa(E), msr(C), vatE*), phenicols (*cat, fexB, poxtA*), and tetracyclines (*tet(45), tet(L), tet(M), tet(O), tet(S), tet(W)*) were identified in the entire collection of 429 isolates (Appendix A). Among these, *eatAv*, *efmA*, *msrC*, *tetL,* and *tet(45)* are associated with multidrug efflux pumps and other transporters, which may potentially lead to resistance against multiple antimicrobials. The number of ARGs identified in each isolate ranged from one to ten. Human *E. faecium* isolates possessed the highest mean number of ARGs per isolate (n = 8) compared to pig and chicken (n = 5, each) and beef cattle isolates (n = 3).

A total of 13 ARGs were identified in the beef cattle *E. faecium* isolate collection, including those imparting resistance to aminoglycosides (*aac(6′)Ii*, and *ant(6)-Ia*), ß-lactams *(pbp5*), fluoroquinolones (*efmA*), MLS (*eatAv*, *lnu(G)*, *vat(E)*, *msr(C)*, and *erm(B)*), and tetracyclines (*tet(45), tet(L)*, *tet(M),* and *tet(S)*) (Figure 2). The most common ARGs observed in beef cattle isolates were plasmid-mediated *aac (6’)-Ii* (extremely high 59/59; 100.0%) and the chromosomally encoded *msr(C)* (extremely high 58/59; 98.3%), *eatAv* (extremely high 45/59; 76.3%), *pbp5* (very high 30/59; 50.8%), and *efmA* (high 21/59; 35.6%), which are responsible for resistance to aminoglycosides, MLS, β-lactams, and fluoroquinolones, respectively. 

Similar to the results obtained for the pig- and chicken-origin *E. faecium* isolates, none of the *E. faecium* isolates from beef cattle were found to possess *gyrA* or *parC* mutations associated with high-level fluoroquinolone resistance. However, some did possess the ABC transporter gene *efmA*, which potentially confers low-level resistance as a fluoroquinolone efflux pump (Figure 2 and Figure 3, Table 1). Similarly, none of the cattle isolates possessed *vanA/vanB* genes encoding resistance to vancomycin (Figure 4) or *liaR/liaS* genes mutations to daptomycin (Figure 5). By comparison, significant proportions of the human sepsis isolates carried a chromosomal quinolone resistance determining region (QRDR) and daptomycin resistance mutations together with vancomycin ARGs (Table 1).

### 2.3. Plasmid Replicons

Overall, 21 plasmid replicon families were observed in the isolate collection. The number of plasmids observed in each isolate ranged from zero to eight, with the highest number of plasmids found in three human isolates (1.0%). Overall, 43 (72.9%) of beef cattle isolates contained at least one plasmid, and 2 isolates harboured six plasmids. A significant number of beef cattle isolates (16/59; 27.1%) did not possess any plasmids compared to human (7/302, 2.3%) and pig isolates (1/60, 1.7%). Plasmid replicon rep1 (broad host range) was predominantly found in beef cattle (33/59; 55.9%) and pig isolates (30/60; 50.0%), but of comparatively lower prevalence in human isolates (15/302; 5.0%), while rep2 (narrow host range) was more commonly found in human (238/302 78.8%) and pig isolates (24/60; 40%) but was present in a much smaller proportion (5/59; 8.5%) of beef cattle isolates. Highly diverse sets of plasmids were identified among the *E. faecium* isolates from human sepsis cases (Figure 6 and Appendix A).

### 2.4. Agreement between Plasmid Replicons and ARG Content

The presence of *vanA* (found only in isolates from humans) was mostly linked to three plasmid replicons: rep2 (98.5%; 64/65), repUS15 (95.4%; 62/65), and rep11a (81.5%; 53/65). Most *E. faecium* isolates carrying the streptogramin resistance gene *vatE* (found only in isolates from animals) were associated with the plasmid replicons rep2 (80%; 4/5) and rep17 (60%; 3/5) (Figure 7). When plasmid replicons and *E. faecium* ARGs known to be plasmid-mediated were cross tabulated and compared according to isolate source (beef cattle vs. human), clear differences in matches were apparent between the two host sources, apart from repUS15, which was highly prevalent in both groups of isolates and often associated with *aac(6’)-Ii*, *erm(B)*, and *tet(M)* ARGs (Figure 8). 

## 3. Discussion

Enterococci are commonly found as commensals in the gastrointestinal tract of animals and humans, but some species, such as *E. faecium*, can cause serious nosocomial infections, including endocarditis, sepsis, and urinary tract infections in humans [20,32]. Regular monitoring and testing of AMR in *E. faecium* isolated from both animal production and human hospital settings can provide important information about the evolution and spread of resistance and the relative impact of food and/or environmental transmission compared to direct human-to-human contact. Through detailed whole bacterial genome bioinformatic analyses, this study focused on AMR, plasmid, and virulence factor profiles identified in *E. faecium* isolated from the gastrointestinal tracts of healthy slaughter-age beef cattle from a single feedlot in Australia and compared these with previous data established for similar pig [33] and meat chicken isolates [28] together with a large collection of clinical isolates from cases of sepsis in humans in Australia [5]. 

The study had three major findings. First, phylogenetic analysis confirmed that the *E. faecium* isolates from healthy slaughter-age beef cattle, pig, and meat chicken sources (found in Clades 1–2b) were genetically distinct from most human clinical isolates, which formed a monophyletic clade (Clade 2ba). Second, none of the animal-origin isolates from beef cattle, pig, and meat chicken sources possessed ARGs or mutations responsible for high-MIC-level resistance to three critically important antimicrobials (ciprofloxacin, vancomycin, and daptomycin) typically present in human isolates. Third, a small proportion (23/302; 7.6%) of human clinical isolates were found clustered with some of the animal isolates (including some belonging to the same ST) within Clades 1–2b, but none of these human isolates possessed ARGs to critically important antimicrobials. These results confirm that, as previously determined for Australian pig and poultry *E. faecium* isolates [33], beef cattle from this particular feedlot also represent an extremely low risk of introducing resistant *E. faecium* (or their ARGs) to humans via the food chain or through the environment with corresponding minimal impact to human health. Evidence for this conclusion can be found in the limited crossover of animal vs. human STs (the majority of beef cattle isolates were located in Clade 1) and the differences in virulence genes and ARG arrays observed, particularly to the critically important antimicrobials (fluoroquinolones, daptomycin, vancomycin). Furthermore, far more plasmid replicons were observed in the *E. faecium* isolated from humans compared to beef cattle isolates. 

The MLST analysis of *E. faecium* isolates from beef cattle in this study revealed significant diversity in their STs with most located in Clade 1. Only three primarily beef-cattle-origin STs (ST22, ST32, and ST327) contained either one or two *E. faecium* isolates obtained from human sepsis cases. This is in agreement with a previous report that clinical *E. faecium* isolates causing infections in humans tend to belong to different sub-lineages than those found in animals, food, and the environment [20]. For example, *E. faecium* ST17, a clonal lineage responsible for hospital-acquired infections throughout the globe, was only identified among human isolates in the present study (where it represented 12.6% of the Australian human *E. faecium* isolate collection). Compared to other strains in the collection, ST17 *E. faecium* were more likely to exhibit high-level resistance to aminoglycosides, beta-lactams, fluoroquinolones, and tetracycline, which correlated with the possession of *aac (6’)-Ii*, *tet (L),* and *tet (M)* genes, as well as mutations in *gyrA*, *parC,* and *pbp5*, as documented in a previous international study [34]. Additionally, the vancomycin resistance gene *vanB* was detected in the hospital-derived ST17 human isolates (3/38; 7.9%). This study found that the human isolates carrying the *van* operon were not clustered in the same evolutionary clade as the beef, chicken, and pig isolates. However, the ST22 and ST32 isolates shared the same ARG types, including plasmid-mediated genes responsible for resistance to aminoglycosides (*aac(6’)-I*); the chromosomally encoded ampicillin resistance mutation (*pbp5*); low-level fluoroquinolone resistance (*efmA*); and macrolide, lincosamide, and streptogramin ARGs (*msr(C)* and *eatAv* in ST327 isolates). It has been previously reported that ST32 and ST22 share the same ARG profiles, yet they are not typical clones associated with vancomycin resistance nor do they have point mutations in *gyrA* and *parC* responsible for high-level fluoroquinolone resistance, indicating a limited human health risk [35,36,37]. Both ST22 and ST32 are categorized as commensal isolates and are part of CC328, which is not associated to any of the established clonal complexes of colonizing or hospital-adapted strains [37]. 

Enterococci are rapidly evolving to become resistant to various classes of antimicrobials, including fluoroquinolones, glycopeptides, lipopeptides, oxazolidinones, and streptogramins [38,39,40]. Within Australia, the streptogramin pristinamycin is also rated of critical importance and grouped with the aforementioned antimicrobial classes in the ASTAG importance ratings [41]. The use of virginiamycin as a feed additive in food-producing animals is believed to contribute to the development of quinupristin/dalfopristin resistance, which are sometimes used as a last resort treatment for vancomycin-resistant *E. faecium* infections in humans [42]. The current study showed that resistance to streptogramins in beef cattle *E. faecium* isolates was comparatively lower compared to human and pig isolates. Some beef *E. faecium* isolates carried the *ermB* gene (3/59; 5.1%), which encodes an enzyme responsible for methylating the 23S rRNA component of the bacterial ribosome, whilst nearly all contained the *msr(C)* gene (58/59; 98.3%), which encodes an efflux pump, resulting in resistance to macrolides and streptogramin B (i.e., quinupristin), and most contained the *eatAV* gene which encodes an ABC transporter and encodes resistance to lincosamide, pleuromutlins, and streptogramin A [43]. Furthermore, one isolate (1.7%) from beef and pig samples and four isolates from meat chickens (50%) were found to harbor the *vatE* gene responsible for streptogramin A (i.e., dalfopristin) resistance. Possession of either *ermB* or *eatAV* with *msr(C)* genes is associated with resistance to quinupristin–dalfopristin, a combination of streptogramin A and B [44]. Whilst a recent survey found that 19.5% of Australian feedlots reported using virginiamycin [45], the present study found that chicken and pig isolates had higher levels of streptogramin resistance genes compared to beef cattle isolates. High prevalence of *msr(C)* in the beef cattle isolates may also be related to the common use of macrolides as first-line treatments for bovine respiratory disease in Australian feedlots [46]. As virginiamycin has not been used in the Australian pig and poultry industries for many years, the higher prevalence of resistance could be related to the co-selection of enterococci resistant to quinupristin–dalfopristin as a result of the historic use of virginiamycin in poultry and current use of tylosin in pigs [33,47]. 

In recent decades, the vancomycin resistance gene *vanA* has been reported in *E. faecium* isolated from animals and animal-derived food products [48]. *VanA*-positive enterococci were isolated from food animals in England in 1993 [49]. Since then, it has been found among food animals worldwide, causing significant public health concerns [50,51,52]. The detection of vancomycin-resistant *E. faecium* has been shown to be associated with the use of avoparcin for growth promotion in food animals [53,54]. However, after vancomycin-resistant enterococci emerged, avoparcin was withdrawn from the market in various countries [55]. In this study, the vancomycin resistance genes *vanA* and *vanB* were found in 21.5% and 28.5% of human isolates, respectively. However, no vancomycin ARGs were found in any of the beef, chicken, and pig isolates. In another study in Australia using phenotypic methods, enterococci isolated from cattle were sensitive to a number of medically important antimicrobials, such as daptomycin, linezolid, tigecycline, and vancomycin [27]. The findings of this study provide evidence that *E. faecium* isolated from Australian healthy slaughter-age beef cattle, pigs, and meat chickens are mutually exclusive from the source of the circulating vancomycin-resistant strains of clinical significance in the public health sector [5]. 

Daptomycin is another antimicrobial used as a last-resort antimicrobial to treat vancomycin-resistant enterococci [40]. However, there have been increasing reports of vancomycin-resistant enterococci (VRE) developing resistance to daptomycin [56]. In this study, none of the *E. faecium* isolated from food-producing animals harboured known resistance genes for daptomycin. However, 23 (7.6%) of the human isolates showed mutations in the genes responsible for daptomycin resistance (*liaR* and *liaS*). Contrary to this, some studies have shown that daptomycin resistance does not always result from *liaFSR* mutations. Mutations in chromosomal genes *cls* and *gdpD*, which encode cardiolipin synthase and glycerophosphoryl diester phosphodiesterase, respectively, have been linked to daptomycin resistance [40]. In this study, it was found that only one human isolate had a mutation in the *cls* gene. In our previously published study, we detected daptomycin resistance in *E. faecium* isolates obtained from both entry (1/9; 11.1%) and exit (21/117; 17.9%) beef cattle faecal samples using a phenotypic method, but the underlying mechanism behind this resistance could not be determined [29]. These findings suggest that there may be other, as yet unknown, pathways responsible for daptomycin resistance in animal-origin enterococci [57,58], including within Australia [28]. 

The prevalence of fluoroquinolone resistance among *E. faecium* isolates is increasing globally, posing a significant threat to public health. In recent years, efflux pumps have played a crucial role in the MDR status of various bacteria [59]. In this study, the *efmA* gene efflux pump, which encodes resistance to fluoroquinolones and macrolides, was observed in *E. faecium* isolated from beef (35.6%), chicken (100%), human (92.4%), and pig (56.7%) samples. Efflux pumps are known to contribute to intrinsic and acquired resistance to antimicrobials used to treat infectious diseases [60]. However, fluoroquinolone resistance characterized by high MIC values is commonly associated with mutations in topoisomerase IV (*parC*) and DNA gyrase (*gyrA*), as previously reported [61]. In this study, no fluoroquinolone-resistance-causing mutations were identified in *E. faecium* isolates obtained from beef cattle. 

The potential for AMR to be transferred to other bacteria exists through several means, with plasmids serving as one of the primary mechanisms for this transfer to occur [62]. Plasmids that contain AMR determinants play an important role in horizontal gene transfer as they often harbour other mobile genetic elements such a transposons, integrons and insertional sequences within their structures [63]. In this study, a previously published plasmid classification scheme was utilized to investigate the possible contribution of specific plasmid families to AMR in *Enterococcus* spp. [13]. Among the human sepsis isolates, *vanA* and *vatE* genes, responsible for resistance to vancomycin and quinupristin-dalfopristin, respectively, were mostly associated with narrow host range rep-2 plasmid replicons. By comparison, the most commonly observed plasmids in *E. faecium* isolates from beef cattle were repUS15 (67.8%) and broad host range rep1 (55.9%), with only a relatively minor proportion carrying rep2 (8.5%). Other international studies have also shown the strong association between carriage of rep2 plasmid replicons and resistance to glycopeptides in enterococci isolated from both humans and food-producing animals [64,65]. In summary, plasmid replicon identification and their associations with critically important ARGs has also shown that Australian beef cattle *E. faecium* isolates (together with pig and meat chicken isolates) pose limited risk to human health from horizontal gene transfer.

Enterococci can carry various virulence genes, including those involved in biofilm formation, adhesion, and invasion of host cells [66]. This study identified a number of virulence genes in the genomes of sequenced *E. faecium* isolates, with the *acm* and *efaAfm* genes being the most commonly detected among all isolates regardless of source and *espfm* only detected in human isolates. Both *acm* and *efaAfm* were prevalent in over 95% of *E. faecium* isolates from Australian beef cattle, similar to the results obtained for chicken and pig isolates. In human *E. faecium* isolates, the virulence genes *espfm* (18.2%) and *hylEfm* (27.1%) were also commonly observed in addition to *acm* and *efaAfm*. The *hylEfm* gene, which is responsible for intestinal colonization, has been detected in ampicillin-resistant *E. faecium* ST17 and vancomycin-resistant *E. faecium* (VREF) strains in hospitals worldwide [67,68,69]. This study shows that the type, resistance, and virulence characteristics of *E. faecium* found in food-producing animals differ from those found in human isolates. 

This study has some limitations that should be taken into account. One major limitation is that the *E. faecium* isolates from meat chickens, pigs, and humans were obtained from a universal database as secondary data. Second, the beef cattle isolates were obtained as part of longitudinal study comparing entry and exit samples from a single feedlot, which may not be sufficient to generalize the results to the wider population. Ideally, this study could now be replicated in a wider range of feedlots. Third, the *E. faecium* isolates from food animals obtained during surveillance were compared with pathogenic isolates from human sepsis cases, and future studies could also examine human carriage isolates. Finally, due to the use of short-read sequencing data, it was not possible to determine the AMR profile and virulence factor composition of each plasmid replicon. In the future, the scope of this study could be expanded to include a range of farms and geographical locations.

## 4. Materials and Methods

### 4.1. Genomic Analysis 

The *E. faecium* isolates from beef cattle were whole-genome sequenced and assembled using a previously described method (BioProject PRJNA879912) [29]. The obtained data were subsequently compared to three previous studies that reported WGS of gastrointestinal *E. faecium* isolates from healthy pigs (accession number PRJNA639902) [33] and meat chickens (accession number PRJNA524396) [28] from slaughterhouses as well as isolates from human sepsis cases (accession number PRJNA562414) [5] from Australia. Assembled sequences with less than 30× coverage and less than 25,000 SNPs were excluded from further analysis. The isolates that met the criteria were from beef cattle (n = 59), pigs (n = 60), meat chickens (n = 8), and human sepsis cases (n = 302). The genetic relationships between isolates were examined using single-nucleotide polymorphisms (SNPs) from cleaned WGS reads that were mapped to an *E. faecium* complete genome (NCBI Assembly Accession: CP003583.1). The software Snippy v4.6.0 (https://github.com/tseemann/snippy, (accessed on 21 September 2022) was used to call core SNPs, i.e., SNPs that can be determined in all isolates. A maximum likelihood (ML) tree was constructed with RAxML v8.2.10 using the model GTRCAT, and a rapid bootstrap analysis with 100 bootstraps for the best-scoring ML tree was undertaken [70]. This was followed by recombination removal using ClonalFrameML v1.12 [71]. Multilocus sequence typing (MLST) was undertaken using MLST 2.0 [72]. The final phylogenetic tree and heat map were manipulated with iTOL (https://itol.embl.de/, accessed on 23 October 2022) for display [73]. A heat map illustrating the presence or absence of each trait for each isolate was created to assess all data elements for all isolates.

### 4.2. Comparative Analysis of Virulence Genes, AMR Genes and Plasmids 

The detection of virulence genes was carried out using VirulenceFinder 2.0 [74,75]. To identify antimicrobial resistance genes (ARGs), we used ResFinder 4.0 [76] and, to further pinpoint the chromosomal point mutations associated with AMR, we used PointFinder [77]. The ResFinder web server (www.genomicepidemiology.org, accessed on 12 October 2022) was used to identify acquired ARGs in the WGS data, using a threshold of 98.0% identity (ID). ARGs were also predicted using the Antibiotic Resistance Genes Database (ARDB) and the Comprehensive Antibiotic Resistance Database (CARD) in addition to ResFinder [78]. To detect plasmid replicons, PlasmidFinder was used, with minimum identity and coverage of 95% and 60% parameters, respectively [79]. 

### 4.3. Statistical Analysis 

Categorical measured traits including ARGs, the presence of plasmid replicons, and virulence factors were converted into a numerical code, with 1 indicating presence and 0 indicating absence. Proportions of AMR, MDR, ARGs, plasmid replicons or virulence factors were calculated using R Statistical Package version 4.0.0. Resistance profiles were categorised as MDR if the isolate exhibited resistance to one or more antimicrobials in three or more antimicrobial classes [80]. AMR and ARG frequencies were described as rare: <0.1%; very low: 0.1 to 1.0%; low: >1 to 10.0%; moderate: >10.0 to 20.0%; high: >20.0 to 50.0%; very high: >50.0 to 70.0%; and extremely high: >70.0% [81].

## 5. Conclusions

The continued use of antimicrobial agents in health care and agriculture settings inevitably leads to the emergence of AMR. This study compared the prevalence of virulence genes, ARGs, and plasmid replicon types identified among *E. faecium* isolates obtained from beef cattle at a single feedlot with those obtained from different animal hosts (pigs and meat chickens) and humans within Australia. Beef cattle, meat chicken, and pig-source *E. faecium* isolates were found to be of limited risk and largely unrelated to human-source isolates. The strict regulation of antimicrobials used in food animals is likely the reason for the low level of AMR found in food animals in Australia. The higher levels of AMR observed in human isolates may be because they are clinical isolates exposed to more antimicrobials than the surveillance isolates from food animals included in this study. This research approach helps understand the general trend and spread of AMR *E. faecium* in different sectors. 

## Figures and Tables

**Figure 1 antibiotics-12-01122-f001:**
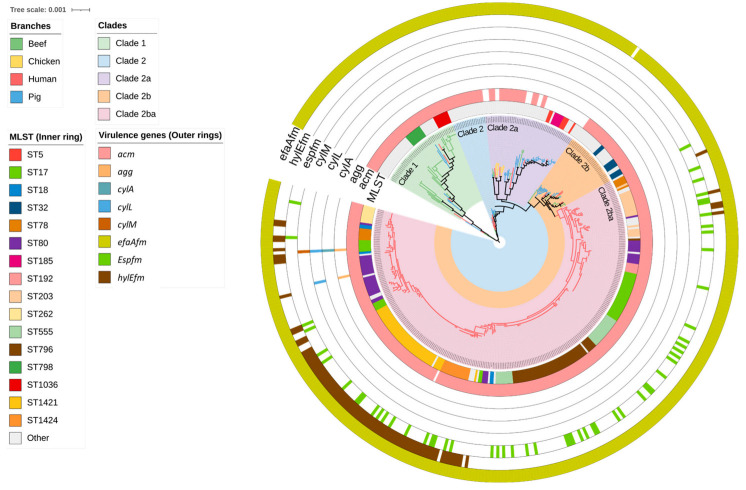
A mid-point rooted, maximum-likelihood phylogenetic tree constructed based on analysis of single-nucleotide polymorphisms (SNPs) of the core SNPs of 429 *Enterococcus faecium* genomes isolated from beef cattle (n = 59), meat chicken (n = 8), pig (n = 60), and human (n = 302) sources. The branch colour indicates the source of the isolates. Clades are coloured according to the legend. The sequence type of the isolates (inner ring) and type of virulence gene (outer rings) are annotated according to the legend.

**Figure 2 antibiotics-12-01122-f002:**
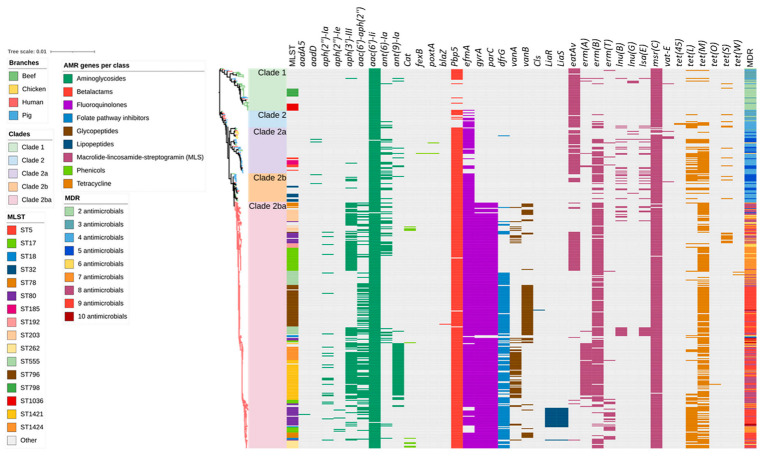
Antimicrobial resistance gene (ARG) profiles clustered on the basis of Figure 1’s SNP-based phylogenetic tree composed of 429 *Enterococcus faecium* genomes isolated from beef cattle (n = 59), chicken (n = 8), pig (n = 60), and human (n = 302) sources. Branches are coloured based on source followed by clade according to the legend. The remaining columns indicate (1) isolate sequence type, (2) ARGs detected in the isolate, and (3) multidrug resistance profile of the isolate. The detected ARGs are clustered according to their respective antimicrobial classes as follows: aminoglycoside (9 ARGs shown in dark green), amphenicols (3 ARGs shown in light green), β-lactams (2 ARGs shown in red), fluoroquinolones (3 ARGs shown in purple), folate synthesis inhibitors (1 ARG shown in light blue), glycopeptides (2 ARGs shown in dark brown), lipopeptides (3 ARGs shown in dark blue), macrolide/lincosamide and/or streptogramins (8 ARGs shown in magenta), and tetracyclines (6 ARGs shown in light brown).

**Figure 3 antibiotics-12-01122-f003:**
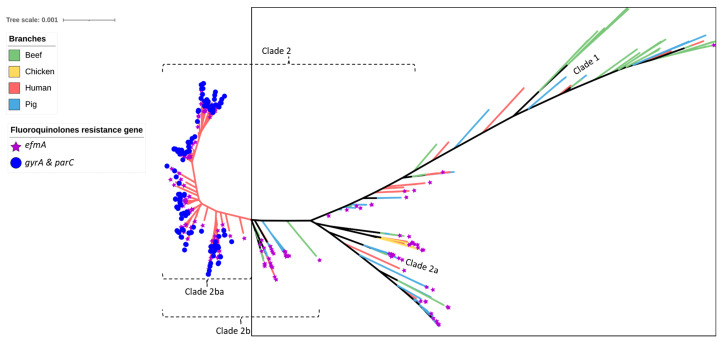
SNP-based phylogeny of 429 *Enterococcus faecium* isolates isolated from beef cattle (green branches, n = 59), meat chickens (orange branches, n = 8), pigs (blue branches, n = 60), and humans (red branches, n = 302). Purple stars indicate the isolates shown to contain the *efmA* efflux pump gene (imparting resistance to fluoroquinolones), whereas the blue circles indicate the isolates shown to contain *gyrA* and *parC* point mutations (imparting high-level resistance to fluoroquinolones). All beef, meat chicken, and pig isolates (encompassed within the square box) contained the efflux pump gene only.

**Figure 4 antibiotics-12-01122-f004:**
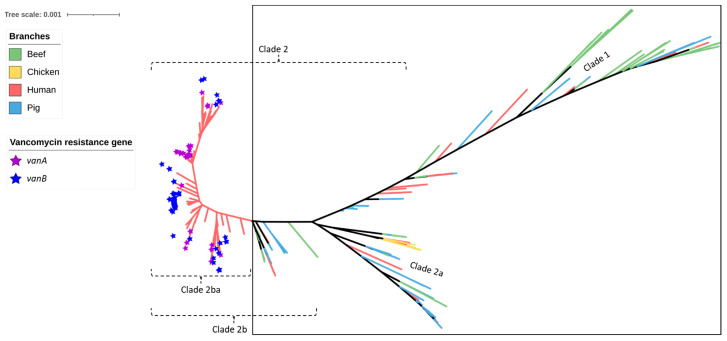
SNP-based phylogeny of 429 *Enterococcus faecium* isolates obtained from beef cattle (green branches, n = 59), meat chickens (orange branches, n = 8), pigs (blue branches, n = 60), and humans (red branches, n = 302). Purple stars indicate the isolates shown to contain *vanA* and blue stars indicate the isolates shown to contain *vanB* (both genes imparting resistance to vancomycin). None of the beef cattle, meat chicken, and pig isolates (encompassed within the square box) possessed *vanA* or *vanB* vancomycin resistance genes.

**Figure 5 antibiotics-12-01122-f005:**
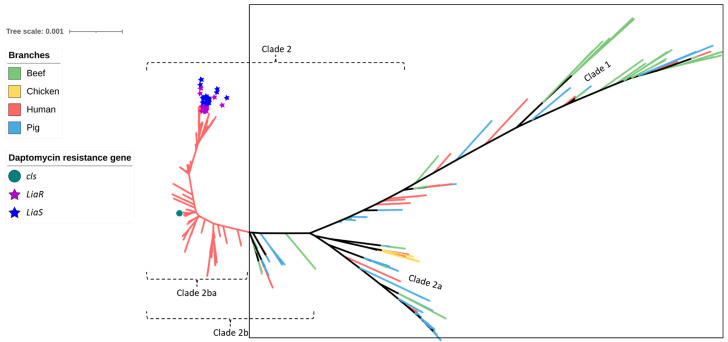
SNP-based phylogeny of 429 *Enterococcus faecium* isolates obtained from beef cattle (green branches, n = 59), meat chickens (orange branches, n = 8), pigs (blue branches, n = 60), and humans (red branches, n = 302). The green circle indicates the isolate shown to contain *cls,* the purple stars indicate the isolates shown to contain *liaR,* and the blue stars indicate the isolates shown to contain *liaS* mutations imparting resistance to daptomycin. None of the beef cattle, meat chicken, and pig isolates (encompassed within the square box) possessed these gene mutations.

**Figure 6 antibiotics-12-01122-f006:**
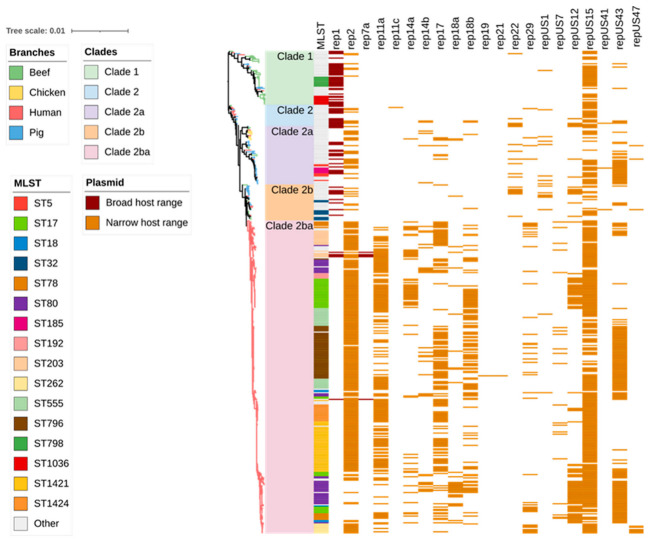
Plasmid replicon profiles clustered on the basis of Figure 1’s SNP-based phylogenetic tree composed of 429 *Enterococcus faecium* genomes isolated from beef cattle (n = 59), meat chickens (n = 8), pigs (n = 60), and humans (n = 302). Branches are coloured based on source followed by clade according to the legend. The remaining columns indicate isolate sequence type and plasmid replicon detected in the isolate. The detected plasmids are clustered according to their respective types as follows: broad host range (2 plasmid replicons shown in dark brown) and narrow host range (19 plasmid replicons shown in light brown).

**Figure 7 antibiotics-12-01122-f007:**
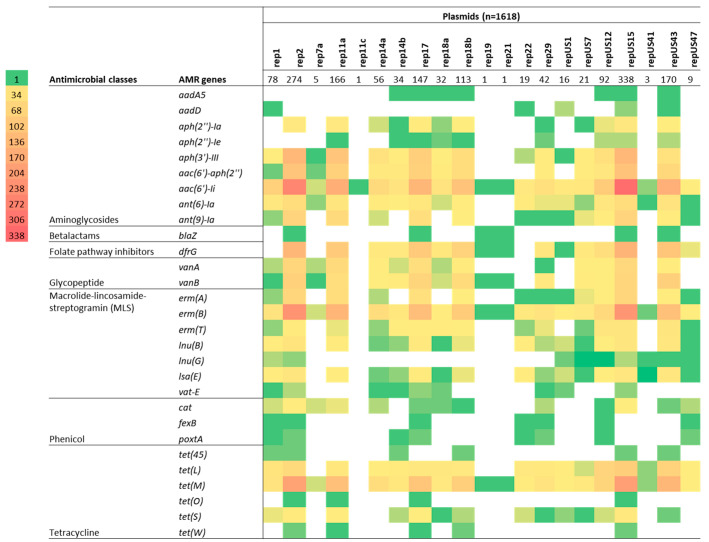
Cross-tabulation heat map showing the degree of correlation between plasmid replicon types and the presence of an antimicrobial resistance gene (ARG) for 429 isolates of *Enterococcus faecium* isolated from beef cattle (n = 59), meat chickens (n = 8), pigs (n = 60), and humans (n = 302). Plasmid replicons are listed horizontally (total number identified in the 429 isolates is also indicated), whereas the ARGs are listed vertically in their classes. The colour strips indicate the number of isolates (in multiples of 34) exhibiting each particular plasmid/ARG match.

**Figure 8 antibiotics-12-01122-f008:**
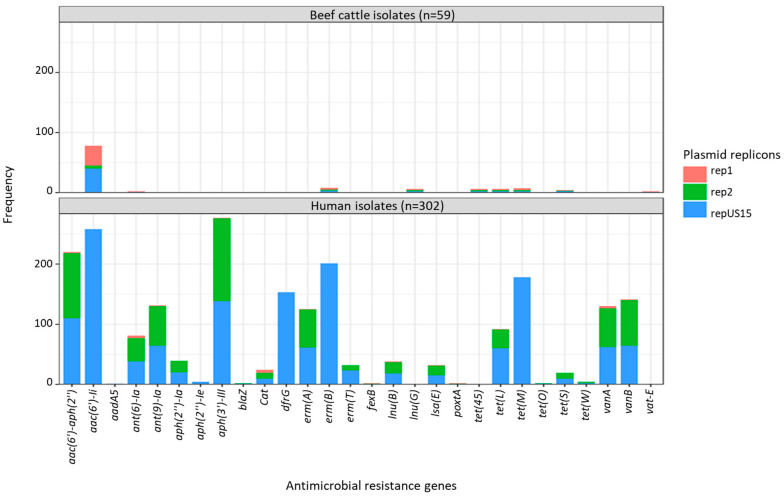
Cross-tabulation of plasmid-mediated antimicrobial resistance genes and plasmid replicons in *Enterococcus faecium* isolated from beef cattle (n = 59) and humans (n = 302). rep1—broad host range shown in pink; rep2 and repUS15—narrow host range shown in green and blue. The ST22 and ST32 isolates, which were detected in beef cattle, pigs, and humans, were clustered together in the same clade. The analysis of ARGs showed that the ST22 and ST32 isolates had similar chromosomal genes for antimicrobial resistance, such as *efmA*, *msr(C),* and *pbp5*, in addition to *aac(6’)-I*, which is more likely to be plasmid-mediated. The pig isolate possessed additional ARGs linked to macrolide and tetracycline resistance. The ST327 isolate obtained from humans and beef cattle were clustered within Clade 1 and carried the chromosomal *eatAv* gene, conferring resistance to lincosamides, streptogramin As, and pleuromutilins in addition to *msr(C)*. Regardless of the presence of the plasmid replicon repUS15, the ARG content in the beef isolates of ST22 and ST32 remained unchanged (Table 2).

**Table 1 antibiotics-12-01122-t001:** Prevalence of antimicrobial resistance genes identified in *E. faecium* isolated from beef cattle, meat chickens, pigs, and humans in Australia.

Antimicrobial Classes	ARGs	Prevalence (%)			
		Beef (n = 59)	Chicken (n = 8)	Human (n = 302)	Pig (n = 60)
Fluoroquinolones	*efmA*	21 (35.6)	8 (100)	279 (92.4)	34 (56.7%)
	*gyrA*	-	-	273 (90.4)	-
	*parC*	-	-	273 (90.4)	-
Glycopeptides	*vanA*	-	-	65 (21.5)	-
	*vanB*	-	-	86 (28.5)	-
Lipopeptides	*cls*	-	-	1 (0.3)	-
	*liaR*	-	-	23 (7.6)	-
	*liaS*	-	-	23 (7.6)	-
Streptogramin A	*eatAv*	45 (76.3)	7 (87.5)	67(22.2)	48 (80.0)
	*lsa(E)*	-	-	17 (5.6)	19 (31.7)
	*VatE*	1 (1.7)	4 (50)	-	1 (1.7)
Streptogramins B	*ermA*	-	-	63 (20.9)	4 (6.7)
	*ermB*	3 (5.1)	-	226 (74.8)	50 (83.3)
	*ermT*	-	-	25 (8.3)	4 (6.7)
	*msrC*	58 (98.3)	6(75.0)	298(98.7)	59 (98.3)

**Table 2 antibiotics-12-01122-t002:** Prevalence of antimicrobial resistance genes, plasmid replicons, and virulence genes in beef cattle, pig, and human *Enterococcus faecium* isolates belonging to the same STs.

Clade	MLST	Isolate (n = 16)	Source of Isolate	No of SNPs	WGS Coverage	Antimicrobial Resistance Genes	Plasmid Replicons	Virulence Genes
2b	22	19MLAP075	Beef cattle	4579	62.1	*aac(6’)-Ii*, *efmA*, *msr(C)*, *pbp5*	rep18b, repUS15	*acm*, *efaAfm*
		SRR10041104	Human	4332	33.5	*aac(6’)-Ii*, *efmA*, *msr(C)*, *pbp5*		*acm*, *efaAfm*
		SRR10041117		3627	28.5	*aac(6’)-Ii*, *efmA*, *msr(C)*, *pbp5*		*acm*, *efaAfm*
		SRR12031114	Pig	2076	26.8	*aac(6’)-Ii*, *efmA*, *erm(B)*, *lnu(G)*, *msr(C)*, *pbp5*, *tet(L)*, *tet(M)*	rep1, rep2, repUS1, repUS41	
		19MLAP293	Beef cattle	4048	55.5	*aac(6’)-Ii*, *efmA*, *msr(C)*, *pbp5*	repUS15	*acm*, *efaAfm*
	32	19MLAP294		3954	55.6	*aac(6’)-Ii*, *efmA*, *msr(C)*, *pbp5*	repUS15	*acm*, *efaAfm*
		19MLAP303		4495	71.8	*aac(6’)-Ii*, *efmA*, *msr(C)*, *pbp5*		*acm*, *efaAfm*
		19MLAP357		4221	50.4	*aac(6’)-Ii*, *efmA*, *msr(C)*, *pbp5*	repUS15	*acm*, *efaAfm*
		19MLAP364		4452	68.0	*aac(6’)-Ii*, *efmA*, *msr(C)*, *pbp5*		*acm*, *efaAfm*
		19MLAP366		4201	81.5	*aac(6’)-Ii*, *efmA*, *msr(C)*, *pbp5*	repUS15	*acm*, *efaAfm*
		19MLAP382		5123	77.7	*aac(6’)-Ii*, *efmA*, *msr(C)*, *pbp5*		*acm*, *efaAfm*
		19MLAP385		4366	50.7	*aac(6’)-Ii*, *efmA*, *msr(C)*, *pbp5*		*acm*, *efaAfm*
		SRR10040901	Human	4544	28.5	*aac(6’)-Ii*, *efmA*, *msr(C)*, *pbp5*		*acm*, *efaAfm*
1	327	19MLAP257	Beef cattle	57,889	44.1	*aac(6’)-Ii*, *eatAv*, *msr(C)*		*acm*, *efaAfm*
		19MLAP391		59,188	64.1	*aac(6’)-Ii*, *eatAv*, *msr(C)*		*acm*, *efaAfm*
		SRR10041173	Human	56,950	32.0	*aac(6’)-Ii*, *eatAv*, *msr(C)*		*acm*, *efaAfm*

## Data Availability

WGS reads are available in the SRA under BioProject beef (PRJNA879912), pig (PRJNA639902), meat chicken (PRJNA524396) and human (PRJNA562414) isolates.

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
