# Peer review of "Phylogeny, Virulence, and Antimicrobial Resistance Gene Profiles of Enterococcus faecium Isolated from Australian Feedlot Cattle and Their Significance to Public and Environmental Health"

_antibiotics, 2023, doi:10.3390/antibiotics12071122_

Round 1

Reviewer 1 Report

There is an omission in line 124 that must be addressed. (The authors mentioned virulence factors in 2 animal meat groups but only included the meat from the cattle, the other animal was not mentioned 

English requires some minor editing for the correction of omissions

Author Response

Thank you!

Reviewer 2 Report

The manuscript is well written and I appreciate the team for this comprehensive work. Wet lab part is minimal in your study as you have isolated only 59 isolates of E. faecium from feedlot cattle. The phylogenetic analysis of AMR, Virulence, MLST and Plasmid replicon highly relied on the previous reports.  Minor grammatical error should be taken care.

Author Response

Thank you!

Please find the attached document.

Reviewer 3 Report

Manuscript is well written. There may be need in future to replicate study in other locations in various Australian provinces. 

Very good quality, except for minor punctuations.

Author Response

Thank you!

Please find the attached document

Reviewer 4 Report

The research carried out is very interesting and with obvious scientific results. The paper could be presented more concisely. They are repetitive expressions. E.g. in line 42 "important causes of hospital-acquired infections in humans" and in line 47 "especially relevant to hospital-acquired infections".

Author Response

(The authors gave the same response as above.)

Reviewer 5 Report

Introduction

Line 31:  Please mention Enterococcus species or strain name used as probiotics.

Results

Line 321 and 330: ([44])    (45) …….check for parentheses.

Materials and Methods

Line 419: The authors may include a sub-section citing the criteria behind selection of species. The authors need to state the reason for including a limited number of chicken samples (n=8).

Line 452: Please write down the software, packages (if using STATA or R software etc.) used to plot the heat map or dendrograms as in figure 6 and 7.

Line 441: As the presence of genes might not reflect the clinical resistance, and to establish that, further analysis is required. If the authors have analysed isolates using phenotypic or automated systems for antimicrobial resistance, the same may be included in the manuscript as supplementary data. 

General Comments:

The study was carried out using the advanced and latest tools for identification of virulence and resistance genes among isolates. The findings like the existence of a limited crossover of animal vs human STs, differences in virulence gene and ARG array between the critically important antimicrobials (fluoroquinolones, daptomycin, vancomycin) make the manuscript interesting. The results may become more comparable and justifiable with the use of phenotypic/automated (eg. VITEK2) methods of antimicrobial susceptibility. For further improvement of the manuscript, the same may be included.

The language is good with minor grammatical errors

Author Response

(The authors gave the same response as above.)
